# Clinical and Epidemiologic Features of Viral Gastroenteritis in Hospitalized Children: An 11-Year Surveillance in Palermo (Sicily)

**DOI:** 10.3390/v15010041

**Published:** 2022-12-22

**Authors:** Emanuele Amodio, Simona De Grazia, Dario Genovese, Floriana Bonura, Chiara Filizzolo, Antonella Collura, Francesca Di Bernardo, Giovanni M. Giammanco

**Affiliations:** 1Dipartimento di Promozione della Salute, Materno-Infantile, di Medicina Interna e Specialistica di Eccellenza “G. D’Alessandro”, Università degli Studi di Palermo, Via del Vespro 133, 90127 Palermo, Italy; 2Unità Operativa di Microbiologia e Virologia, Ospedale Civico e Di Cristina, ARNAS, 90127 Palermo, Italy

**Keywords:** viral gastroenteritis, children, rotavirus, norovirus, adenovirus, astrovirus, Italy

## Abstract

In order to acquire information regarding viral agents and epidemiologic features of severe paediatric Viral Acute Gastroenteritis (VAGE) across multiple seasons in the pre-rotavirus-vaccine era, the epidemiologic characteristics of VAGE were investigated among paediatric patients hospitalized in a major Sicilian paediatric hospital from 2003 to 2013. Overall, 4725 children were observed and 2355 (49.8%) were diagnosed with a viral infection: 1448 (30.6%) were found positive to rotavirus, 645 (13.7%) to norovirus, 216 (4.6%) to adenovirus, and 46 (0.97%) to astrovirus. Viral infections showed different patterns of hospitalization in terms of age at risk (younger for rotavirus and adenovirus infections), seasonality (increased risk in winter for rotavirus and norovirus), trend over time (reduced risk in 2011–2013 for norovirus and rotavirus) and major diagnostic categories (digestive diseases more frequent in adenovirus and astrovirus but not in norovirus). This study increases general knowledge of VAGE epidemiology and contributes to suggest some a priori diagnostic criteria that could help clinicians to identify and treat viral agents responsible for gastroenteritis in hospital settings.

## 1. Introduction

Acute gastroenteritis is a very common paediatric illness and remains a significant cause of childhood morbidity and mortality worldwide [1]. Acute gastroenteritis can be caused by a broad spectrum of enteric pathogens, but in children, a large proportion of cases are due to enteric viruses.

Children younger than 5 years of age are most susceptible to severe viral acute gastroenteritis (VAGE), and 3–5 billion cases per year of acute gastroenteritis and nearly two million deaths occurred worldwide at the dawn of the third millennium [2]; furthermore, in 2017, the Institute for Health Metrics and Evaluation still estimated 1.5 million deaths for diarrhoeal diseases, mostly in low-income countries [3]. In higher-income economies the cost of gastroenteritis to the community is huge, but often underestimated, if costs to the family, including lost time at work, are not considered besides those linked to hospitalizations and medical care [4].

Despite the previously reported considerations, complete information regarding enteric viral agents and epidemiologic features of severe paediatric VAGE across multiple seasons is lacking in several countries, including developed countries like Italy, among others. Moreover, hospital-based paediatric diarrhoea studies are mostly performed focusing on laboratory detection of viral pathogens, frequently omitting the clinical and epidemiological characteristics of infected children. Viral and epidemiologic data from hospital records would be useful to help explain the reasons leading to the hospitalization of children who suffer VAGE. Over 20 different viruses have been identified as etiologic agents of VAGE [5]; however, rotavirus (Reoviridae family), norovirus (Caliciviridae family), adenovirus (Adenoviridae family), and astrovirus (Astroviridae family) account for most clinical cases. The prevalence of these etiologic agents is influenced by environmental, geographical, or socio-economic factors [6]. Since most VAGE cases are due to rotavirus infections [7], in several regions, including Sicily, rotavirus vaccines have been introduced to reduce VAGE morbidity [8,9,10,11,12].

In order to increase the knowledge on the viral agents involved in VAGE hospitalizations, we have monitored the epidemiologic trends in VAGE among hospitalized paediatric patients, carrying out an 11-year study to investigate the viral aetiologies in the pre-rotavirus-vaccine era and their associated epidemiological and clinical features. The epidemiologic characteristics of the hospitalized children (e.g., demographic and clinical characteristics of the paediatric patients, hospitalization length and diagnosis related group (DRG) weight, and seasonality) were correlated to the different viruses detected as etiologic agents.

## 2. Methods

### 2.1. Study Population and Samples

This is an observational study evaluating the epidemiology of acute gastroenteritis due to four main viral aetiologies in Sicilian children. This study was conducted from 2003 to 2013 in a major Sicilian Hospital (ARNAS Civico Hospital, Palermo), that is also the major paediatric hospital of Palermo and surrounding area (about 1.2 million residents). Children were considered eligible for this study if they were less than 5 years old and presented with at least one of the following gastro-enteric signs or symptoms: vomiting, diarrhoea, abdominal cramps, fever, poor appetite, and dehydration. Those who met the inclusion criteria (N = 5584) were enrolled in this study. For each patient the respective hospital discharge record was linked and information about age, sex, date of hospital admission, date of hospital discharge, residency, nationality, diagnosis related group (DRG, according to Italian Medicare Severity categories) weight, and major diagnostic category were extracted. Coinfections were excluded in order to facilitate the comparison of variables between viral agents.

A total of 858 (15.4%) children were excluded from the analysis: 550 because the hospital discharge record was not linkable, and 308 because a coinfection was present.

Stool samples were collected from patients during the period of hospitalization. Antigenic tests were performed at the Microbiology and Virology Laboratory of the ARNAS Civico hospital of Palermo and biomolecular analyses were carried out at the Microbiology and Virology Laboratory of the Azienda Policlinico Universitario “P. Giaccone” of Palermo. Antigenic tests were performed on fresh stool samples that were frozen and stored at −80 °C until the time of biomolecular assays. All of the stool specimens were tested for rotavirus, norovirus, enteric adenovirus, and astrovirus. In particular, enteric adenovirus antigens were detected in fresh stools by Immunocromatography (ICT) assays (CerTest Rotavirus + Adenovirus, Biotec S.L, Zaragoza, Spain, and VIKIA Rota-Adeno, bioMerieux SA, Lyon, France), whilst molecular assays for the detection of viral RNA were used for rotavirus, norovirus, and astrovirus. Viral RNA was extracted from 140 μL 10% faecal suspensions using the QIAamp Viral RNA kit according to the manufacturer’s instructions (QIAGEN, Hilden, Germany). Astrovirus RNA was detected by a conventional Reverse transcription (RT)-PCR [13]. For rotavirus and norovirus, reverse transcription of RNA was carried out using random primers, as previously described [14], and viral genome was detected by specific Real-time PCR targeting the NSP3 gene and the ORF1/ORF2 junction, respectively, as previously described [15,16].

### 2.2. Statistical Analysis

Comparison of categorical variables between viral agents was performed with a χ2 test, whereas differences among the numerical variables were analysed by ANOVA or Wilcoxon signed-rank test as appropriate. A multinomial logistic regression analysis was applied to explore factors associated with infection, evaluating the adjusted odds ratio of positivity to each virus compared to the others (other viruses + patients without viral isolation).

Statistical significance was defined as a *p*-value of <0.05. The descriptive statistics were performed using R statistical software version 3.6.1.

## 3. Results

Overall, 4725 children were observed during the study period, and for 2355 (49.8%) of them, a viral infection was detected (Table 1). The M:F ratio was 1.33 (with 57.1% of males) and the median age of recruited patients was 1.38 years. Most of hospital admissions (1369; 29%) were in the winter season, with a sharp increase in the 2011–2013 period (47.6% out of all patients included in this study). The median hospital stay was 4 days and 43% of all hospitalizations were coded as due to diseases of the digestive system.

In Table 2, the epidemiological factors and clinical features in hospitalized children according to virus detection is reported. Among 4725 patients under study, 1448 (30.6%) were found positive to rotavirus, 645 (13.7%) to norovirus, 216 (4.6%) to adenovirus, and 46 (0.97%) to astrovirus. Statistically significant differences were found between groups for age (*p* < 0.001), nationality (*p* < 0.001), hospital admission season (*p* < 0.001), period of hospital admission (*p* < 0.001), length of hospital stay (*p* = 0.048), DRG weight (*p* = 0.003), and diagnosis according to major diagnostic categories (*p* < 0.001).

Table 3 shows a multinomial logistic regression analysis that was carried out comparing the risk factors for each viral infection with patients infected with other viruses or with no viral detection. Patients with adenovirus detection had a lower risk of infection in winter and spring, and higher odds of entering the hospital at a younger age and for diseases of the respiratory or digestive system.

Patients with astrovirus detection were older and more frequently suffered from digestive and nutritional/metabolic diseases.

Children with norovirus infection had a higher risk of being hospitalized in autumn or winter, whereas their hospitalization risk decreased in 2011–2013.

Subjects infected by rotavirus had a higher risk of being younger, affected by digestive and nutritional/metabolic diseases, hospitalized in winter or spring, and with a lower DRG weight. Their hospitalization risk decreased statistically significantly in 2011–2013.

Figure 1 depicts seasonality of hospitalized paediatric acute gastroenteritis stratified by causative virus and surveillance period. Rotavirus was most prevalent in spring, but its season also included winter and summer; norovirus prevailed in late autumn and winter, and adenovirus had a peak during the summer/fall in 2011–2013.

## 4. Discussion

The present study provides comprehensive information on the epidemiologic and clinical features associated with four enteric viruses most commonly found in children. The simultaneous evaluation of these viruses should be considered of strategic importance since it provides information on aetiology-specific features, which are relevant in studies of viral behaviour and useful in clinical practice. According to the international literature, we have found that rotavirus infections represented the first infective etiologic cause of viral gastroenteritis requiring hospitalization among children prior to the implementation of rotavirus vaccination, accounting for 30.8% of all patients with acute gastroenteritis. The primary role of rotavirus in acute gastroenteritis in children is confirmed by almost all previous studies that have been conducted in the pre-vaccine era [17,18,19]. Several authors also reported a significant reduction in the hospitalizations due to rotavirus after rotavirus vaccine introduction [8,9,12].

Historically, noroviruses have been recognized as the second-most frequent etiologic agent of viral gastroenteritis in children [18,19,20], reporting a high burden of infection also in Sicily where food and water-borne transmission have been demonstrated to play an important role [21,22].

In our study, adenovirus and astrovirus accounted for a smaller fraction (about 5%) of all patients, although other authors suggested that they could have a major role in determining acute gastroenteritis [22,23,24].

Intriguingly, the four VAGE agents had different presentation with regard to age (rotavirus and adenovirus infected significantly younger children), main diagnosis/associated health problems, and seasonality. A relatively high percentage of VAGE children were categorized as hospitalized for respiratory diseases (DRG 04) or problems of ear, nose, mouth, and throat (DRG 03). Although this could seem a classification inaccuracy, it is well documented that in clinical settings gastrointestinal symptoms are often accompanied by signs of respiratory infections. Respiratory symptoms, such as sore throat, nasal discharge, sneezing, and coughing have been described in up to 27% of gastroenteritis patients [25,26,27]. In particular, a very high risk of respiratory diseases (22.5% among our infected patients) was found in patients with norovirus infections, although this result was not significant in the multivariable test, suggesting that the high incidence of norovirus infections during the winter season, when influenza-like illnesses are also more frequent, could be confounding the risk analysis. Otherwise, patients with adenovirus infections had a significant higher risk for respiratory diseases also, after adjusting for confounders, confirming results from the international literature [28,29].

With respect to seasonality according to other authors, we observed that the number of rotavirus infections appeared to be significantly higher from February to May [17,30,31], whereas norovirus infections were detected more frequently in winter months [32,33]. In our study adenovirus peaked in early fall in 2011–2013, while a large majority of studies report peaks in summer [32]. However, some other authors have found differences in seasonality, suggesting that adenovirus circulation may be dependent on the area studied and climatic conditions [34].

This study has some limitations including a single centre setting and lack of some clinical information (e.g., vaccination status, patient’s symptoms and signs). Moreover, only some viral agents have been investigated, excluding minor enteric viruses and bacterial and parasitic microorganisms responsible for paediatric acute gastroenteritis. Finally, a possible increase in the surveillance system sensitivity in the latest period under analysis (2011–2013) cannot be ruled out, although we are confident that the increase in the number of cases did not generate significant selection bias and, thus, the accuracy of the risk evaluation reported in the present study was maintained over the time periods analysed.

Despite these possible limitations, the present study enriches the general knowledge on four main viruses implicated in VAGE, suggesting different clinical presentations and features. In particular, this study confirms that viral agents are a common cause of hospital admission in previously healthy children during the first years of life, accounting for at least 50% of all acute gastroenteritis. Rotaviruses and noroviruses are the most common pathogenic agents, with the former being characterized by well-defined digestive or metabolic (dehydration) health problems. Moreover, a clear seasonality was evident for rotavirus (late winter/spring), norovirus (winter), and adenovirus (late summer/fall). The high burden attributable to rotavirus and norovirus confirms the need for high-coverage rotavirus vaccination (introduced in Sicily in 2013) and for the development of additional vaccines for high-burden viruses (norovirus) or emerging ones (adenovirus). In the meantime, deepening the insight on the epidemiology of these viruses remains of value for those several countries where rotavirus vaccine coverage is still very low. Moreover, also in countries with high vaccination coverage, a percentage of vaccinated children could suffer from rotavirus breakthrough infections, since the vaccine is not 100% effective [35].

Future prospective studies are needed to validate the clinical importance of our results, contributing to identify VAGE diagnostic characteristics that can improve patient management. Finally, the epidemiological data obtained in this study, including seasonality, age, and severity of illness, will represent a useful baseline to evaluate any changing pattern of presentation of common enteric viruses after two years of the COVID-19 pandemic.

## Figures and Tables

**Figure 1 viruses-15-00041-f001:**
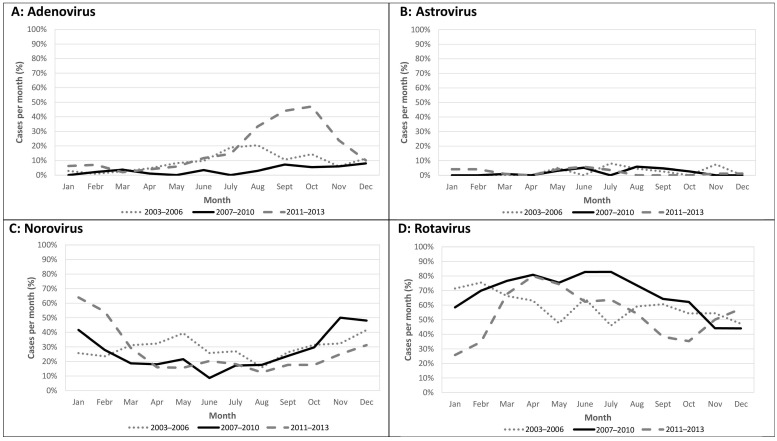
Seasonality of paediatric acute gastroenteritis (number of cases per month/total viral cases per month *100) stratified by causative virus ((**A**): Adenovirus, (**B**): Astrovirus, (**C**): Norovirus, (**D**): Rotavirus) and study period (2003–2006, 2007–2010, 2011–2013).

**Table 1 viruses-15-00041-t001:** Epidemiological factors and clinical features in hospitalized children (n = 4725) with viral acute gastroenteritis.

	All Patients (4725)
Number of detections, n (%)	2355 (49.8%)
Males, n (%)	2701 (57.2%)
Age in years, median (IQR)	1.38 (0.64–2.55)
Urban residency, n (%)	2767 (58.6%)
Italian nationality, n (%)	4650 (98.4%)
Admission season (2003–2013), n (%)	
- Summer	1069 (22.6%)
- Autumn	1027 (21.7%)
- Winter	1369 (29%)
- Spring	1260 (26.7%)
Admission year, n (%)	
- 2003–2006	1209 (25.6%)
- 2007–2010	1265 (26.7%)
- 2011–2013	2251 (47.6%)
Hospital stay in days, median (IQR)	4 (2–6)
DRG weight in units, mean (SD)	0.65 (0.30)
Major DRG, n (%)	
- 03 Diseases of the Ear, Nose, Mouth, and Throat	417 (8.8%)
- 04 Diseases of the Respiratory System	751 (15.9%)
- 06 Diseases of the Digestive System	2029 (43%)
- 10 Diseases of the Endocrine, Nutritional, Meta-bolic	720 (15.1%)
- Others	806 (17.1%)

**Table 2 viruses-15-00041-t002:** Epidemiological factors and clinical features in hospitalized children according to virus detection.

	Adenovirus	Astrovirus	Norovirus	Rotavirus	No Isolation	*p*-Value
Number of isolations, n (by row %)	216 (4.6%)	46 (0.97%)	645 (13.7%)	1448 (30.6%)	2370 (50.2%)	-
Males, n (%)	120 (55.6%)	27 (58.7%)	385 (59.7%)	796 (55%)	1373 (57.9%)	0.25
Age in years, median (IQR)	1.4 (0.7–2.5)	2.7 (1.4–5.4)	1.2 (0.6–2.3)	1.4 (0.7–2.3)	1.4 (0.6–2.8)	<0.001
Urban residency, n (by column %)	131 (60.6%)	24 (52.2%)	374 (58%)	858 (59.3%)	1398 (59%)	0.84
Italian nationality, n (by column %)	204 (94.4%)	45 (97.8%)	641 (99.4%)	1439 (99.4%)	2321 (97.9%)	<0.001
Admission season, n (by column %)						
- Summer	78 (36.1%)	12 (26.1%)	82 (12.7%)	258 (17.8%)	639 (27%)	<0.001
- Autumn	82 (38.0%)	17 (37%)	134 (20.8%)	456 (31.5%)	621 (26.2%)	
- Winter	24 (11.1%)	8 (17.4%)	154 (23.9%)	244 (16.9%)	539 (22.7%)	
- Spring	32 (14.8%)	9 (19.6%)	275 (42.6%)	490 (33.8%)	571 (24.1%)	
Admission period, n (prevalence/1000)						
- 2003–2006	51 (42.2)	14 (11.6)	202 (167.1)	418 (345.7)	524 (433.4)	<0.001
- 2007–2010	19 (15.0)	11 (8.7)	181 (143.1)	519 (410.3)	535 (422.9)	
- 2011–2013	146 (64.9)	21 (9.3)	262 (116.4)	511 (227.0)	1311 (582.4)	
Hospital stay in days, median (IQR)	3 (2–5)	3 (2–5)	4 (2–6)	4 (2–6)	3 (2–6)	0.048
DRG weight in units, mean (SD)	0.62 (0.19)	0.61 (0.19)	0.66 (0.32)	0.63 (0.27)	0.67 (0.32)	0.003
Major DRG, n (by column %)						
- 03 Diseases of the Ear, Nose, Mouth, and Throat	17 (7.8%)	3 (6.5%)	78 (12.1%)	115 (7.9%)	204 (8.6%)	<0.001
- 04 Diseases of the Respiratory System	28 (12.9%)	5 (10.9%)	145 (22.5%)	220 (15.2%)	353 (14.9%)	
- 06 Diseases of the Digestive System	125 (57.9%)	24 (52.2%)	209 (32.4%)	671 (46.3%)	1000 (42.2%)	
- 10 Diseases of the Endocrine, Nutritional, Metabolic	28 (13.0%)	12 (26.1%)	101 (15.7%)	244 (16.9%)	335 (14.1%)	
- Others	18 (8.3%)	2 (4.3%)	112 (17.4%)	198 (13.7%)	476 (20.1%)	

**Table 3 viruses-15-00041-t003:** Multivariable analysis on risk factors for having an infection with the indicated virus vs. other virus detections/no detection.

	Adenovirus OR (95% CI)	Astrovirus OR (95% CI)	Norovirus OR (95% CI)	Rotavirus OR (95% CI)
Sex, ref. females	0.92 (0.68–1.24)	1.06 (0.59–1.93)	1.09 (0.91–1.3)	0.91 (0.8–1.05)
Age in years, per year increment	**0.87 (0.8–0.94) ^c^**	**1.15 (1.05–1.25) ^c^**	0.98 (0.94–1.02)	**0.91 (0.88–0.94) ^c^**
Hospital stay, in days	0.96 (0.91–1.01)	1.01 (0.94–1.08)	0.99 (0.97–1.01)	0.99 (0.98–1.01)
DRG weight, in units	0.65 (0.26–1.61)	0.4 (0.04–3.9)	1.01 (0.74–1.38)	**0.71 (0.53–0.95) ^a^**
Admission season, ref. Summer				
- Autumn	1.16 (0.82–1.65)	0.77 (0.31–1.91)	**2.32 (1.73–3.12) ^c^**	**1.28 (1.03–1.59) ^a^**
- Winter	**0.28 (0.17–0.48) ^c^**	1 (0.41–2.44)	**3.65 (2.76–4.83) ^c^**	**2.34 (1.92–2.85) ^c^**
- Spring	**0.38 (0.24–0.6) ^c^**	1.55 (0.73–3.3)	**1.76 (1.31–2.37) ^c^**	**2 (1.65–2.43) ^c^**
Admission period, ref. 2003–2006				
- 2007–2010	**0.02 (0–0.13) ^c^**	0.78 (0.34–1.79)	0.96 (0.75–1.22)	1.18 (0.98–1.42)
- 2011–2013	1.15 (0.8–1.64)	0.61 (0.29–1.28)	**0.55 (0.44–0.68) ^c^**	**0.49 (0.41–0.58) ^c^**
Major DRG, ref. Other major DRGs				
- 03 Diseases of the Ear, Nose, Mouth, and Throat	1.93 (0.85–4.37)	2.47 (0.34–17.9)	1.42 (0.97–2.06)	1.01 (0.73–1.39)
- 04 Diseases of the Respiratory System	**2.41 (1.22–4.75) ^a^**	3.61 (0.67–19.39)	1.35 (1–1.82)	1.13 (0.88–1.46)
- 06 Diseases of the Digestive System	**3.63 (2.06–6.41) ^c^**	**4.98 (1.14–21.75) ^a^**	0.94 (0.71–1.23)	**1.57 (1.28–1.93) ^c^**
- 10 Diseases of the Endocrine, Nutritional, Metabolic	1.89 (0.92–3.84)	**5.85 (1.23–27.77) ^a^**	1.1 (0.79–1.52)	**1.45 (1.12–1.87) ^a^**

^a^ *p*-value < 0.05; ^c^
*p*-value < 0.001.

## Data Availability

The data are available under reasonable request to the corresponding authors.

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
