# Peer review of "Clinical and Epidemiologic Features of Viral Gastroenteritis in Hospitalized Children: An 11-Year Surveillance in Palermo (Sicily)"

_viruses, 2022, doi:10.3390/v15010041_

Round 1

Reviewer 1 Report

This manuscript summarized the epidemiology of four enteric pathogens (rotavirus, norovirus, adenovirus, astrovirus) using the hospital-based dataset collected during 11 year surveillance in Sicily before the introduction of rotavirus vaccines. During this surveillance authors collected information from 4725 children and detected these enteric viruses from 2355 patients. The data showed that, as shown in other countries, rotavirus was the most predominant enteric viruses in children, followed by norovirus, with similar seasonality trend, in Sicily. Overall, the data is well summarized and consistent with other studies conducted in different locations. Minor comments/suggestions are listed below to improve the manuscript.

1. Please provide figure legend for Fig.1. This graph shows the number of VAGE cases in y axis and months in x axis but please clarify if it shows an average or total number of cases in 11 years. Ideally, I would suggest to plot the number of cases per year or per 3-4 years (2003-2006, 2007-2010, 2011-2013) as number of cases are very different by years (very high in the last three years) and there may be variations in seasonality by years.

2. In 2011-2013, majority of cases showed no isolation (55%, n = 1311). Do you have any ideas what could be the causative agents for this sudden increase of VAGE cases in this period?

3. There was a higher number of adenovirus cases in 2011-2013 as compared with previous years (146 vs 51 or 19) but there is no statistical significance in Table 3. Likewise, the number of norovirus cases increased in 2011-2013 (262) as compared with previous years (202, 181), but multivariable analysis indicated an opposite relationship (significantly lower risk of norovirus infection in 2011-2013). Could you please explain why this discrepancy happen? If you looked at the detection rate per year, norovirus infection could be lowered in 2011-2013 due to the large number of unknown VAGE cases, but it does not always mean that the number of norovirus infection itself was decreased in this period. Unless the surveillance system was drastically changed and more patients were recruited in this period in 2011-2013, I cannot see why you could say that the norovirus infection risk is low (or no difference in adenovirus infection risk) in 2011-2013 while the actual number of cases increased.

4. Related to the comment #3 above, why there is no significant difference in age in norovirus cases in Table 3? Table 2 shows that age of norovirus cases is lowest (median and range) among the four viruses. 

Author Response

Reviewer #1:

Dear Reviewer,

Thank you for revising our manuscript and for your suggestions and comments that allow us to improve the quality of the manuscript.

Below you will find a point-by-point answer to each raised question. We hope this improved version of the manuscript can be considered suitable for publication on Viruses.

Question: Please provide figure legend for Fig.1. This graph shows the number of VAGE cases in y axis and months in x axis but please clarify if it shows an average or total number of cases in 11 years. Ideally, I would suggest to plot the number of cases per year or per 3-4 years (2003-2006, 2007-2010, 2011-2013) as number of cases are very different by years (very high in the last three years) and there may be variations in seasonality by years.

Answer: Figure 1 was replaced and, according to you suggestion, a plot showing the rate of cases per 3-4 years was produced in order to show variations in seasonality by years. We also tried to plot number cases per year per each causative agent but the figure was too confused. A new legend was added clarifying that VAGE cases are reported as number of cases per month/total viral cases per month *100. The text of the results section was also modified accordingly.

Question: In 2011-2013, majority of cases showed no isolation (55%, n = 1311). Do you have any ideas what could be the causative agents for this sudden increase of VAGE cases in this period?

Answer: We apologize for representing this result in a confused way. In 2011-2013 there was an increase in the analyzed samples due, probably, to a better performance of our surveillance system. However, in 2011-2013 the percentage of samples without isolation was quite similar to the previous periods (1,311/2,251, 58.2% vs. 524/1,209, 43.3% in 2003-2006 and 535/1,265, 42.3% in 2007-2010). According to the previous considerations and in order to increase clarity of result representation, data reported in table 2 were modified and prevalence*1,000 analyzed samples were reported.  

Question: There was a higher number of adenovirus cases in 2011-2013 as compared with previous years (146 vs 51 or 19) but there is no statistical significance in Table 3. Likewise, the number of norovirus cases increased in 2011-2013 (262) as compared with previous years (202, 181), but multivariable analysis indicated an opposite relationship (significantly lower risk of norovirus infection in 2011-2013). Could you please explain why this discrepancy happen? If you looked at the detection rate per year, norovirus infection could be lowered in 2011-2013 due to the large number of unknown VAGE cases, but it does not always mean that the number of norovirus infection itself was decreased in this period. Unless the surveillance system was drastically changed and more patients were recruited in this period in 2011-2013, I cannot see why you could say that the norovirus infection risk is low (or no difference in adenovirus infection risk) in 2011-2013 while the actual number of cases increased.

Answer: Thank you very much for giving us the opportunity to clarify this very important point. Unfortunately, the multivariable models can only evaluate the odds of being positive for a causative agent according to a precise variable (in our case period). As you rightly state, in 2011-2013 the number of norovirus cases increased but the odds of being positive for norovirus in 2011-2013 decreased significantly because in 2011-2013 we had a higher number of evaluated samples (for instance prevalence of norovirus occurrence decreased from 167.1/1,000 samples in 2003-2006 to 143.1/1,000 samples in 2007-2010 and 116.4/1,000 samples in 2011-2013). Accordingly, also in the considered model there was a significant reduced risk for isolating norovirus in 2011-2013. A similar situation happened for adenovirus although in this case we observed a not significant increase in prevalence over time (42.2/1,000 samples in 2003-2006, 15.0/1,000 samples in 2007-2010 and 64.9/1,000 samples in 2011-2013). In this latter case it is possible that the small amount of positive cases could have decreased the precision of the estimates with a consequent not significant result. Unfortunately, we are convinced that a change in the surveillance system sensitivity can not be excluded and that, at least a part of the increased number of isolations observed in 2011-2013 for almost all causative agents, could be attributed to this change. Accordingly, we added this statement to the limits of the study in the Discussion section.

Question:  Related to the comment #3 above, why there is no significant difference in age in norovirus cases in Table 3? Table 2 shows that age of norovirus cases is lowest (median and range) among the four viruses.

Answer: As for you, we were surprised about this result. However, we verified the rightness of the model and we confirm the obtained result, thus we think that for norovirus the contribute of age could not be large enough for determining a statistical significance when considering other preeminent variables as seasonality or period.  

Reviewer 2 Report

In the manuscript, the authors reported some statistics related to viral acute gastroenteritis in Italy.  They showed some characteristics of rotavirus, norovirus, adenovirus, and astrovirus infections in the pre-introduction era of rotavirus vaccination.  However, the importance of results in the manuscript was unclear at present in the post-introduction era of rotavirus vaccination.  The authors should discuss about the importance of publishing historical data before acceptance in Viruses. 

Author Response

Reviewer #2:

Dear Reviewer,

thank you for revising our manuscript and for your appreciation in the regards of the latter.

Below you will find the answer to your question.

Question: However, the importance of results in the manuscript was unclear at present in the post-introduction era of rotavirus vaccination. The authors should discuss about the importance of publishing historical data before acceptance in Viruses.

Answer: Thank you very much for these considerations that allow us to clarify our point of view. We think that our study can be of value for healthcare despite the introduction of rotavirus vaccination for several reasons. The first reason is that unfortunately rotavirus disease is still a very important health problem in a large majority of the countries due to the low vaccination coverage. Moreover, also in countries with quite high vaccination coverage a relatively important number of children will acquire rotavirus infection due to the fact that vaccine is not 100% effective, and a percentage of vaccinated children could suffer from rotavirus breakthrough infections. According to these considerations, we are still confident that the epidemiology of rotavirus and other gastrointestinal viruses can be of great impact for scientific purposes. All these considerations and a new reference were added to the manuscript in the Discussion section.     

Reviewer 3 Report

The study from Amodio and colleagues is an important addition to our current understanding of pediatric viral enteric disease based on hospital surveillance. The findings are not completely novel as they are similar to those reported in previous studies, but nevertheless are beneficious to clinicians in order to achieve the most accurate diagnostic of viral gastroenteritis. Also, the findings generated in this study reinforce why was needed the introduction of a rotavirus vaccine globally.

 Pg 2 Ln 60: Authors should rephrase the sentence “to answer some unsolved questions”, to avoid an ambitious approach.

Author Response

Reviewer #3:

Dear Reviewer,

Thank you for revising our manuscript and for appreciation of our results.

As recommended, we have changed the sentence “to answer some unsolved questions” and rephrased it in “in order to increase the knowledge on the viral agents involved in VAGE hospitalizations”.